# Confirmatory factor analysis of the Tinnitus Impact Questionnaire using data from patients seeking help for tinnitus alone or tinnitus combined with hyperacusis

Hashir Aazh [1,2], Brian C. J. Moore [1,3], Mercede Erfanian [1,4]*

1 Hashir International Specialist Clinics & Research Institute for Misophonia, Tinnitus and Hyperacusis Ltd, London, United Kingdom, 2 Audiology Department, Royal Surrey NHS Foundation Trust, Guildford, United Kingdom, 3 Cambridge Hearing Group, Department of Psychology, University of Cambridge, Cambridge, United Kingdom, 4 Institute for Environmental Design and Engineering, The Bartlett, University College London, London, United Kingdom

* ucbqmer@ucl.ac.uk

**Data Availability Statement:** Data cannot be shared publicly. Data contain potentially identifying or sensitive patient information. In addition, the

## Abstract

A confirmatory factor analysis (CFA) of the Tinnitus Impact Questionnaire (TIQ) was performed. In contrast to commonly used tinnitus questionnaires, the TIQ is intended solely to assess the impact of tinnitus by not including items related to hearing loss or tinnitus loudness. This was a psychometric study based on a retrospective cross-sectional analysis of clinical data. Data were available for 155 new patients who had attended a tinnitus and hyperacusis clinic in the UK within a five-month period and had completed the TIQ. The mean age was 54 years (standard deviation = 14 years). The TIQ demonstrated good internal consistency, with Cronbach's α = 0.84 and McDonald's ω = 0.89. CFA showed that two items of the TIQ had low factor loadings for both one-factor and two-factor models and their scores showed low correlations with scores for other items. Bi-factor analysis gave a better fit, indicated by a relative chi-square ($\chi^2$) of 18.5, a Root-Mean Square Error of Approximation (RMSEA) of 0.103, a Comparative Fit Index (CFI) of 0.97, a Tucker Lewis Index (TLI) of 0.92, and a Standardized Root-Mean Residual (SPMR) of 0.038. Total TIQ scores were moderately correlated with scores for the Visual Analogue Scale of effect of tinnitus on life and the Screening for Anxiety and Depression-Tinnitus questionnaire, supporting the convergent validity of the TIQ. The TIQ score was not correlated with the pure-tone average hearing threshold, indicating discriminant validity. A multiple-causes multiple-indicator (MIMIC) model showed no influences of age, gender or hearing status on TIQ item scores. The TIQ is an internally consistent tool. CFA suggests a bi-factor model with sufficient unidimensionality to support the use of the overall TIQ score for assessing the impact of tinnitus. TIQ scores are distinct from the impact of hearing impairment among patients who have tinnitus combined with hearing loss.

data are owned by a third-party organization (Tinnitus and Hyperacusis Therapy Specialist Clinic (THTSC), Audiology Department at the Royal Surrey NHS Foundation Trust (RSFT)). Analysis of the data was approved by the Southwest-Cornwall and Plymouth Research Ethics Committee and the Research and Development department at the RSFT (Project ID: 182924). Data are available from the Institutional Data Access / Ethics Committee (contact via info@hashirtinnitusclinic.com) for researchers who meet the criteria for access to confidential data.

**Funding:** The author(s) received no specific funding for this work.

**Competing interests:** The authors have declared that no competing interests exist.

## Introduction

According to the World Health Organization's International Classification of Functioning for disability and health (WHO ICF), the severity of symptoms of a condition is different from the impact of the condition on the patient's life [1]. The Tinnitus Impact Questionnaire (TIQ) was developed as a clinical tool for assessing the impact of tinnitus on a patient's life [2]. Consistent with the WHO ICF model, it is important to note that the loudness of tinnitus (which reflects the severity of the symptom) is not the same construct as its impact. Aazh, Lammaing and Moore [3] reported that despite a significant correlation between tinnitus loudness as measured using the Visual Analogue Scale (VAS) and tinnitus impact as measured via the Tinnitus Handicap Inventory (THI) [4], tinnitus loudness did not predict the impact of tinnitus when the effect of other important factors (tinnitus annoyance, anxiety and depression) were taken into account in a linear multiple-regression model. Most interventions for tinnitus focus on reducing its impact as opposed to reducing its loudness, so outcome measures should be able to assess the impact of tinnitus on the patient's life separately from its loudness [5,6]. The TIQ solely focuses on assessing the impact of tinnitus and does not include any items that are related to its loudness. This is different from another commonly used questionnaire, the Tinnitus Functional Index (TFI) [7], which has some items asking about the severity of symptoms, for example, asking "How STRONG or LOUD was your tinnitus?" and "What percentage of your time awake were you consciously AWARE of your tinnitus?"

Another important factor is the misconception that tinnitus interferes with hearing and the understanding of speech. This has led the authors of most commonly used questionnaires, such as the THI, the TFI, the Tinnitus Reaction Questionnaire (TRQ) [8], the Tinnitus Primary Functions Questionnaire (TPFQ) [9], the Tinnitus Handicap Questionnaire (THQ) [10], and the Tinnitus Questionnaire (TQ) [11], to include items that assess the impact of tinnitus on hearing. Hearing loss is highly prevalent among tinnitus patients and tinnitus loudness tends to increase with increasing severity of the hearing loss [12]. However, recent studies show that tinnitus does not interfere with the perception of external sounds and individuals with tinnitus can detect and discriminate sounds as well as people with matched hearing thresholds in a control group with no tinnitus [13]. It seems likely that when a person has tinnitus combined with hearing loss, difficulties in hearing are caused primarily by the hearing loss itself, not by the tinnitus. However, such individuals may mistakenly blame their tinnitus for their hearing difficulties [14]. Questionnaires that include items related to the impact of tinnitus on hearing mix the construct of "hearing loss impact" with "tinnitus impact". It is plausible that the experience of tinnitus is different among people who also have hearing loss. However, to investigate this, tools are needed that measure the impact of tinnitus separately from the impact of hearing loss. To sum up, in contrast to commonly used tinnitus questionnaires, the TIQ is intended to give a pure assessment of the impact of tinnitus by not including items related to hearing loss or tinnitus loudness. For a full discussion of the implications of mixing different constructs in a tinnitus questionnaire, the reader is referred to a previous paper on the TIQ, which also reported the results of exploratory factor analysis (EFA) of the TIQ [2].

EFA was previously conducted for an earlier version of the TIQ that had nine items [2]. In that version of the TIQ, there were two items related to sleep, comprising: "Delay in falling asleep" and "Difficulty going back to sleep if woken up during the night". The EFA revealed a high correlation between responses to these two items, rendering one of the items redundant. Similarly, scores for an item about 'feeling worried' showed a strong a correlation with scores for items about 'feeling anxious' and 'having low mood'. As a result, it was decided to combine the sleep items so as to create a new sleep item and to remove the item on 'feeling worried' [2].

The resulting seven-item TIQ was found to be a one-factor questionnaire having excellent internal consistency, with a Cronbach's alpha (α) of 0.89.

It is recommended to conduct Confirmatory Factor Analysis (CFA) on a new questionnaire, especially if items are removed during EFA, in order to validate the factor structure and to ensure that the questionnaire is measuring the intended construct [15]. In addition, as recommended by the COSMIN guideline [15] and established literature [16], EFA should always be followed by CFA to validate the latent constructs. In the present study, CFA was conducted using TIQ responses from a new group of tinnitus patients, separate from the group whose responses were used for the EFA.

The aims of the present study were to assess (1) the factor structure of the TIQ using CFA, and (2) the discriminant and convergent validity of the TIQ and its internal consistency.

## Materials and methods

### Ethical approval

The study was registered and approved as a clinical audit by the Quality Governance Department at the Royal Surrey NHS Foundation Trust (RSFT). Patient consent was waived, as this was a retrospective analysis of the available clinical data. Analysis of the data was approved by the Southwest-Cornwall and Plymouth Research Ethics Committee and the Research and Development department at the RSFT (Project ID: 182924). In accordance with best practices in research transparency and to minimize research biases, this study was pre-registered on AsPredicted.org (AsPredicted #157218).

### Study design and patients

This was a retrospective cross-sectional study conducted at the RSFT, Guildford, UK. In a five-month period in 2022 (15 Dec 2021 to 28 April 2022), 200 adults sought help for tinnitus or tinnitus combined with hyperacusis from the Tinnitus and Hyperacusis Therapy Specialist Clinic (THTSC). The authors had no access to information that could identify individual participants during or after data collection, as all data were fully anonymized. Demographic data for the patients, the results of their audiological evaluations and their responses for self-report questionnaires were imported from their records held at the Audiology Department. Complete data were not available for all measures for some patients. As a part of their routine care, a case history was taken to explore each patient's medical history, tinnitus characteristics, laterality of tinnitus, presence of ear pain, ear discharge (otorrhea), balance problems, whether the patient had seen an Ear, Nose, and Throat (ENT) specialist, whether they reported perceiving tinnitus in their dreams, history of exposure to loud noises, family history of autistic spectrum disorders (ASD), history of childhood parental mental illness, family history of tinnitus and/or noise sensitivity, history of mental illness, if they had seen mental health services, and if they had any suicidal ideations. The information gathered in the case history was largely consistent with the recommendations of the British Academy of Audiology [17] and the European School for Interdisciplinary Tinnitus Research (ESITR)–Screening Questionnaire [18], and aimed to help the audiologist to decide if the patient should be referred to other medical or psychiatric specialists. There were some questions outside the BAA or ESITR guidelines that asked about: (1) perceiving tinnitus in dreams (it has been reported that this is associated with a more severe impact of tinnitus) [19], (2) a history of parental mental illness in childhood (this is associated with an increased severity of symptoms of anxiety and depression) [20], and (3) if they have any suicidal ideations. This question was asked only for patients who brought up the topic of self-harm or suicide during the history taking session and/or reported a history of parental mental illness in their childhood and/or had an abnormal score on their mental health self-

report measure (the questionnaire used is detailed under the self-report questionnaires) [21–23].

## Pure tone audiometry

Pure-tone audiograms were measured following the guidelines of the British Society of Audiology [24], but with certain modifications proposed by Aazh and Moore [25] to mitigate any potential discomfort. The severity of hearing loss was assessed based on the pure-tone average (PTA) across the frequencies 0.25, 0.5, 1, 2, and 4 kHz, consistent with the BSA guidelines [24], and classified as Mild (20–40 dB HL), Moderate (41–70 dB HL), Severe (71–95 dB HL), or Profound (over 95 dB HL).

## Self-report questionnaires

All questionnaires were completed as a part of standard care prior to the start of any treatment, at each patient's first visit to the THTSC. Patients completed the online versions of the questionnaires including the TIQ, Screening for Anxiety and Depression in Tinnitus (SAD-T), Sound Sensitivity Symptoms Questionnaire (SSSQ), Hyperacusis Impact Questionnaire (HIQ) and Visual Analogue Scale (VAS) of tinnitus loudness (VAS Tinnitus Loudness), annoyance (VAS Annoyance), and effect on life (VAS Effect on Life), without the involvement of their audiologist. Each questionnaire is described below.

**Tinnitus Impact Questionnaire (TIQ).** The TIQ assesses how often respondents experience a number of problems because of tinnitus, over a two-week period. The seven items used in the TIQ, shown in Table 1, were developed based on the common complaints of tinnitus patients attending the THTSC. The items were chosen to explore the effect of tinnitus on day-to-day activities and mood. Each situation described in the TIQ is assigned a score of 0, 1, 2, or 3, based on the number of days it occurred over a 2-week period: 0–1 days, 2–6 days, 7–10 days, and 11–14 days. Overall scores range from 0 to 21. Cronbach's alpha for the TIQ was reported to be 0.89 [2].

**Screening for Anxiety and Depression in Tinnitus (SAD-T).** The SAD-T includes four items aligned with those in the Physical Health Questionnaire (PHQ-4) [26]. Two items pertain to anxiety and two to depression. The response choices and their scoring are similar to those for the TIQ. Scores for the SAD-T range from 0 to 12, with a score of 4 or higher indicating the presence of symptoms of anxiety and/or depression. The SAD-T has a Cronbach's alpha of 0.91 [27].

**Sound Sensitivity Symptoms Questionnaire (SSSQ).** The SSSQ is a five-item questionnaire designed to assess various sound sensitivity symptoms as described by Tyler, Pienkowski [28], such as loudness hyperacusis, pain or discomfort hyperacusis, annoyance hyperacusis/

**Table 1. Items and response choices of the Tinnitus Impact Questionnaire (TIQ).**

| Over the last 2 weeks, how often would you say the following has occurred because of hearing a sound in your ears or head with no external source (e.g., buzzing, a high-pitched whistle, hissing...)? | | | | |
|---|---|---|---|---|
| 1. Lack of concentration | days | 2–6 days | 7–10 days | 11–14 days |
| 2. Feeling anxious | 0–1 days | 2–6 days | 7–10 days | 11–14 days |
| 3. Sleep difficulties (delay in falling asleep and/or difficulty getting back to sleep if woken up during the night) | 0–1 nights | 2–6 nights | 7–10 nights | 11–14 nights |
| 4. Lack of enjoyment from leisure activities | 0–1 days | 2–6 days | 7–10 days | 11–14 days |
| 5. Inability to perform certain day-to-day activities/tasks | 0–1 days | 2–6 days | 7–10 days | 11–14 days |
| 6. Feeling irritable | 0–1 days | 2–6 days | 7–10 days | 11–14 days |
| 7. Low mood | 0–1 days | 2–6 days | 7–10 days | 11–14 days |

misophonia, and fear hyperacusis, over a two-week period [27]. The response choices and their scoring are similar to those for the TIQ. The total score ranges from 0 to 15, and scores of 5 or more indicate sound tolerance problems. Cronbach's alpha for the SSSQ is 0.87 [27].

**Hyperacusis Impact Questionnaire (HIQ).**   The HIQ is an eight-item questionnaire assessing the impact of hyperacusis on the patient's life. The HIQ asks respondents how often (in number of days in the last 14 days) each of several situations occurred because of certain environmental sounds that seemed too loud to them, but that other people could tolerate well. The response choices and their scoring are similar to those for the TIQ. The overall score ranges from 0 to 24. Scores above 11 indicate a clinically significant impact of hyperacusis [27]. Cronbach's alpha for the HIQ is 0.93 [27].

**Visual Analogue Scale (VAS).**   Tinnitus loudness, annoyance, and impact on the patient's life were assessed using a VAS [29]. To assess tinnitus loudness, patients were asked to rate the loudness of their tinnitus during their waking hours over the past month, 0 indicating the absence of tinnitus and 10 indicating the loudest tinnitus imaginable. The annoyance induced by the tinnitus was assessed by asking patients to rate their average level of annoyance experienced over the past month on a scale ranging from 0 (no annoyance) to 10 (highest annoyance imaginable). To assess the impact of tinnitus, patients were asked to rate the extent to which tinnitus affected their life over the past month on a scale ranging from 0 (no impact) to 10 (extreme impact).

## Data analysis

To investigate the possible role of selection bias [30,31], Mann-Whitney U tests were used to assess the significance of differences between TIQ responders and non-responders in age, gender, total scores for the SAD-T, HIQ, and SSSQ, VAS ratings for tinnitus loudness, annoyance, and the impact on life, and PTA values for the better and worse ears and across ears.

**Structural validity: Factor analysis.**   CFA with the weighted least-squares mean and variance-adjusted estimator (WLSMV) [32] was applied to the data for responders to assess whether the previously suggested one-dimensional factor structure of the TIQ could be confirmed and to assess the goodness of fit of that and other models. For a factor structure to be considered satisfactory, factor loadings were required to be >0.50 for each item.

Three models were used to assess the factor structure of the TIQ scores [33,34]: a one-factor model, a two-factor model, and a bi-factor model. The one-factor model assessed whether all items were primarily associated with a single latent trait. If so, a single score would effectively capture the construct of interest. The two-factor model assessed whether the responses were linked to two separate latent traits, each of which had a certain weighting. The bi-factor model assessed the extent to which the responses could be explained by an overarching primary trait accompanied by smaller independent traits that had only a small influence on the total score [33,34].

Several goodness of fit indices were used to assess the fit of the three models. A close fit is suggested by: a relative chi-square value ($\chi^2$/df) close to 2 [35], a Root-Mean Square Error of Approximation (RMSEA) [36] and a Standardized Root-Mean Residual (SRMR) [37] below 0.05, a Tucker-Lewis Index (TLI) [38] above 0.9 and a Comparative Fit Index (CFI) [36] above 0.95.

**Internal consistency.**   The internal consistency of the TIQ was assessed via Cronbach's alpha ($\alpha$) [39] and McDonald's [40] omega ($\omega$), for which values $\geq$ 0.7 indicate satisfactory consistency. Item-total correlations (ITC) were also calculated for the TIQ. The ITC for a given item is the correlation between scores for that item and scores for the total excluding that item. Values between 0.3 and 0.8 are required [41]. Finally, the value of alpha when each

item was deleted and the polychoric inter-item correlations for categorical responses were calculated.

The item endorsement rate/frequency is a measure of the degree to which respondents give the same response for a given item [42]. For example, if 16% (N = 25) of respondents give a response of 7–10 days for the item about "low mood" (TIQ7), the endorsement rate for that item/response is 16%. The endorsement rate was analysed to assess how well the TIQ items differentiate between patients in terms of the impact of tinnitus. To this end, categorical items were created (0: item was not endorsed (0–1 days), 1: item endorsed at level 1 (2–6 days), 2: items endorsed at level 2 (7–10 days), and 3: item endorsed at level 3 (11–14 days).

**Discriminant and convergent validity.** Discriminant validity was assessed by exploring the correlations between TIQ scores and the scores for measures that are thought to be only weakly related to the impact of tinnitus, namely the PTA of the better ear, the PTA of the worse ear, the PTA averaged across ears and the VAS for tinnitus loudness [12,14,43]. Convergent validity was assessed by exploring the correlation between TIQ scores and scores for other measures that assess constructs that are thought to be related to the impact of tinnitus. These comprised the VAS for tinnitus annoyance and effect on life, hyperacusis as measured via the HIQ and SSSQ, and anxiety and depression as measured via the SAD-T [3,14,44,45]. Spearman correlation coefficients were interpreted via the general rule used within medical statistics, that is 0.3–0.5 as low, 0.51–0.7 as moderate, and 0.71–0.9 as strong [46].

In order to assign categories of mild, moderate, and severe tinnitus impact based on the HIQ, Aazh, Hayes [2] used THI categories as a reference. In their study, for each of the four tinnitus handicap categories of the THI, the cut-off points for the TIQ were established. This led to the following categories: a TIQ score below 5 indicates no impact of tinnitus, a score of 5 or 6 indicates mild impact, a score of 7 or 8 indicates moderate impact, and a score of 9 or more indicates a severe impact. In the current study, the discriminant and convergent validity of these TIQ ordinal categories were evaluated by comparing scores for the SAD-T, HIQ, SSSQ, VAS for tinnitus loudness, annoyance and effect on life, and hearing threshold measures for individuals falling in different tinnitus impact categories, using analysis of variance. The Kruskal-Wallis test was used for multiple-group comparisons of non-normally distributed and unequal-sample groups. Subsequently, Wilcoxon rank-sum tests were conducted to determine the significance of pairwise comparisons between tinnitus impact categories, using Bonferroni correction to account for multiple comparisons.

**Measurement invariance.** The measurement invariance of the TIQ with respect to age, gender and hearing status was assessed using a multiple causes multiple indicator (MIMIC) model [47]. This model allows the regression of items and latent factors onto one or more covariates. Measurement non-invariance in a model may occur when there is a significant direct effect of one or more covariates. Direct effects with values ≤0.36 were considered to have a small magnitude [48]. Age was discretized into categorical values, at intervals of 10 years, and the BSA suggested categories of mild, moderate, severe and profound [24] hearing loss for the better ears, worse ears, and the average across ears were used as independent variables in the MIMIC analysis. The measurement invariance analysis was conducted for each TIQ item rather than the TIQ total score [49].

Statistical analysis was conducted in MATLAB (v. R2022b, Leipzig, Germany - www. mathworks.com) [50] and RStudio (2023.03) [51]. The CFA and the analysis of measurement invariance were conducted in RStudio, using the 'lavaan' package [52], along with the 'Sem-Tools' package designed for categorical data [53]. The 'lavaan' package facilitates CFA by specifying models, estimating parameters, assessing fit, and modifying models for optimal representation of relationships between latent and observed variables. The *p* value for significance was a priori set to *p* < 0.05 for all analyses.

## Results

### Participants and their characteristics

The analysis was based on the data from 155 patients who completed the TIQ out of 200 (response rate of 77.5%). Failure to complete the TIQ occurred mainly due to time constraints in the clinics. There was no significant difference between responders and non-responders for any measures except HIQ scores, VAS for tinnitus annoyance and PTA of the worse ears, which were all worse among the responders (Table 2).

Table 3 shows demographic characteristics of the responders. Most had seen an ENT specialist and most had bilateral tinnitus. About 9% of patients had suicidal ideations. These patients were either already under the care of mental health services or were referred to a psychiatrist following their assessment in audiology.

Audiograms were available for only 73 patients. Some patients who lacked audiograms had undergone hearing assessments at other medical facilities prior to their referral, and these did not need to be repeated for the purpose of treating tinnitus and/or hyperacusis. Of the 73, for the better ear, 47 (64%) participants had no hearing loss, and 23 (32%) and 3 (4%) had mild and moderate hearing loss, respectively. For the worse ear, 30 (41%) had no hearing loss, 32 (44%) had mild loss, 8 (11%) had moderate loss, and 3 (4%) had severe hearing loss. Of the 134 patients who completed the HIQ, 10 (7.5%) exhibited symptoms of hyperacusis. Of the 148 patients who completed the SSSQ, 24 (16%) showed symptoms of sound sensitivity. Of the 118

**Table 2. Comparison of age, gender and scores for the self-report questionnaires between patients who completed the TIQ as a part of their routine clinical assessment (responders) and patients who did not complete the TIQ but completed other measures (non-responders).**

| Measure | TIQ responders Mean (SD) | TIQ non-responders Mean (SD) | z-statistic |
|---|---|---|---|
| **Age** | 54 (14), n = 155 | 534 (15), n = 45 | 0.16 |
| **Gender** | 1.5 (0.5), n = 155 | 1.5 (0.5), n = 45 | -0.3 |
| **SAD-T [a]** | 3.8 (4.0), n = 118 | 2.5 (4.8), n = 11 | 1.6 |
| **SSSQ [b]** | 2.0 (3.0), n = 148 | 2.0 (3.6), n = 20 | 0.9 |
| **HIQ [c]** | 2.9 (5.0), n = 134 | 0.9 (2.8), n = 19 | **2.4*** |
| **VAS (A) [d]** | 6.4 (1.8), n = 154 | 5.9 (2.0), n = 38 | **2.3*** |
| **VAS (L) [e]** | 6.6 (2.6), n = 154 | 6.2 (2.3), n = 38 | 1.2 |
| **VAS (E) [f]** | 6.0 (2.7), n = 154 | 5.9 (2.3), n = 38 | 0.6 |
| **PTA (b) (dB HL) [g]** | 17.5 (10), n = 73 | 20.7 (12), n = 29 | -1.3 |
| **PTA (w) (dB HL) [h]** | 26.0 (17), n = 73 | 17.8 (14), n = 29 | **2.6**** |
| **PTA (a) (dB HL) [i]** | 21.7 (13), n = 73 | 21.1 (17), n = 29 | 0.7 |

[a] Screening for Anxiety and Depression.

[b] Sound Sensitivity Symptoms Questionnaire.

[c] Hyperacusis Impact Questionnaire.

[d] Visual Analogue Scale Annoyance.

[e] Visual Analogue Scale Loudness.

[f] Visual Analogue Scale Effect on life.

[g] Pure Tone audiometry better ear.

[h] Pure Tone audiometry worse ear.

[i] Pure Tone audiometry across ears.

*p <0.05

**0.01

***0.001.

**Table 3. Demographic information for responders based on the data collected via routine history taking (n = 155).**

| Demographic characteristics | n (%) |
|---|---|
| Ear pain | 39 (25%) |
| Ear discharge (Otorrhea) | 3 (2%) |
| Balance problems | 58 (37%) |
| Seen Ear, Nose, and Throat (ENT) specialist | 115 (74%) |
| History of loud noise exposure | 59 (38%) |
| Perceiving tinnitus in their dreams | 9 (6%) |
| Family history of autistic spectrum disorders (ASD) | 0 |
| History of parent mental illness in childhood | 23 (15%) |
| Family history of tinnitus, sensitivity to noise, or hearing impairment | 42 (27%) |
| Bilateral tinnitus | 99 (64%) |
| Unilateral tinnitus | |
| Right ear | 21 (14%) |
| Left ear | 27 (17%) |
| History of mental illness | 68 (44%) |
| Seen mental health services | 43 (28%) |
| Suicidal ideations | |
| Not indicated | 66 (43%) |
| No | 41 (26%) |
| Yes, but no plan | 12 (8%) |
| Yes, with plan | 2 (1%) |

patients who completed the SAD-T, 52 (44%) displayed symptoms related to anxiety and depression.

The TIQ, SSSQ, and SAD-T total scores were all positively skewed, with median scores of 6 (Q1 = 3, Q3 = 11, range from 0 to 21), 0 (Q1 = 0, Q3 = 3, range from 0 to 15), and 2 (Q1 = 0, Q3 = 6, range from 0 to 12), respectively. There was a significant difference in TIQ scores between patients with (mean = 2.8, SD = 5.0) and without (mean = 0.9, SD = 2.9) hyperacusis ($z = 2.4$, $p < 0.05$). Patients who had tinnitus combined with hyperacusis tended to have higher TIQ scores than patients who did not have hyperacusis.

## Factor structure and internal consistency

The TIQ demonstrated good internal consistency, with Cronbach's $\alpha = 0.84$ and $\omega = 0.89$. The measures of goodness of fit for the three CFA models are shown in Table 4. The one-dimensional model gave an inadequate fit. The standardized factor loadings ranged from 0.29 to 0.88, indicating problematic items and an unsuitable solution. The two-factor model also failed

**Table 4. Confirmatory one-factor, two-factor, and bi-factor goodness-of-fit indices for the TIQ (n = 155).** The number of parameters was 21 for the bi-factor model and 14 for the other two models.

| Fit Index | One-factor | Two-factor | Bi-factor |
|---|---|---|---|
| $\chi^2/df$ | 49.4/14 = 3.53 | 91.1/15 = 6.05 | 18.5/7 = 2.64 |
| RMSEA | 0.128 | 0.181 | 0.103 |
| CFI | 0.919 | 0.826 | 0.974 |
| TLI | 0.878 | 0.756 | 0.921 |
| SRMR | 0.067 | 0.206 | 0.038 |

**Table 5. Standardised factor loadings and reliability statistics of the bi-factor model for the TIQ.**

| Item | Label | Median (Q1, Q3) | Loading | Item-total correlation | Alpha if item deleted |
|------|-------|-----------------|---------|------------------------|----------------------|
| TIQ1 | Lack of concentration | 1 (0, 3) | 0.87 | 0.78 | 0.80 |
| TIQ2 | Feeling anxious | 0 (0, 2) | 0.69 | 0.81 | 0.79 |
| TIQ3 | Sleep difficulties (delay in falling sleep and/or difficulty getting back to sleep if woken up during the night) | 1 (0, 3) | 0.58 | 0.52 | 0.86 |
| TIQ4 | Lack of enjoyment from leisure activities | 0 (0, 1) | 0.58 | 0.73 | 0.81 |
| TIQ5 | Inability to perform certain day-to-day activities/tasks | 0 (0, 0) | 0.51 | 0.61 | 0.83 |
| TIQ6 | Feeling irritable | 1 (0, 3) | 0.85 | 0.75 | 0.81 |
| TIQ7 | Low mood | 1 (0, 2) | 0.65 | 0.81 | 0.79 |

to give a good fit. In particular, the relative $\chi^2$ value was 6.07, well above the target value of 2. The standardized factor loadings for the two-factor model ranged from 0.6 to 0.89 for the first factor (including TIQ1, TIQ2, TIQ4, TIQ6 and TIQ7) and were 0.37 and 0.57 for the second factor (including TIQ3 and TIQ5), also indicating a poor fit. The bi-factor model gave a better fit than the other two models, with a relative $\chi^2$ value of 2.64, closely approaching the target value of 2, and yielded factor loadings ranging from 0.51 to 0.87, all of which were >0.5 (Table 5).

Polychoric correlations showed that scores for TIQ3 were poorly correlated with scores for the other items (Table 6). However, as shown in Fig 1, TIQ3 exhibited the highest endorsement frequency for response option 3 (11–14 nights). This means that most responders had a significant problem with their sleep.

## Discriminant and convergent validity of the TIQ

Scores for the SAD-T and VAS of tinnitus effect on life were moderately correlated with TIQ total scores, providing evidence for convergent validity (Table 7). The correlations between TIQ total scores and PTA values of the better ears, worse ears and across ears were very small and were non-significant, indicating discriminant validity. The TIQ total score was weakly correlated with scores for the SSSQ, HIQ, and VAS for tinnitus annoyance and tinnitus loudness, indicating that the construct of tinnitus impact as measured via the TIQ is different from but somewhat related to the constructs measured by these questionnaires. Age and gender were not significantly correlated with the TIQ total score or item scores.

**Table 6. Correlations between TIQ total scores and TIQ item scores.**

| Item | TIQ1 | TIQ2 | TIQ3 | TIQ4 | TIQ5 | TIQ6 | TIQ7 | TIQ (Total) |
|------|------|------|------|------|------|------|------|-------------|
| TIQ1 | 1.00 | 0.55 | 0.31 | 0.53 | 0.39 | 0.53 | 0.50 | **0.77**\*\*\* |
| TIQ2 | 0.55 | 1.00 | 0.21 | 0.60 | 0.36 | 0.49 | 0.79 | **0.80**\*\*\* |
| TIQ3 | 0.31 | 0.21 | 1.00 | 0.13 | 0.21 | 0.37 | 0.23 | **0.52**\*\*\* |
| TIQ4 | 0.53 | 0.60 | 0.13 | 1.00 | 0.43 | 0.38 | 0.63 | **0.73**\*\*\* |
| TIQ5 | 0.39 | 0.36 | 0.21 | 0.43 | 1.00 | 0.42 | 0.39 | **0.61**\*\*\* |
| TIQ6 | 0.53 | 0.49 | 0.37 | 0.38 | 0.42 | 1.00 | 0.52 | **0.75**\*\*\* |
| TIQ7 | 0.50 | 0.79 | 0.23 | 0.63 | 0.39 | 0.52 | 1.00 | **0.81**\*\*\* |

\*p <0.05

\*\*0.01

\*\*\*0.001.

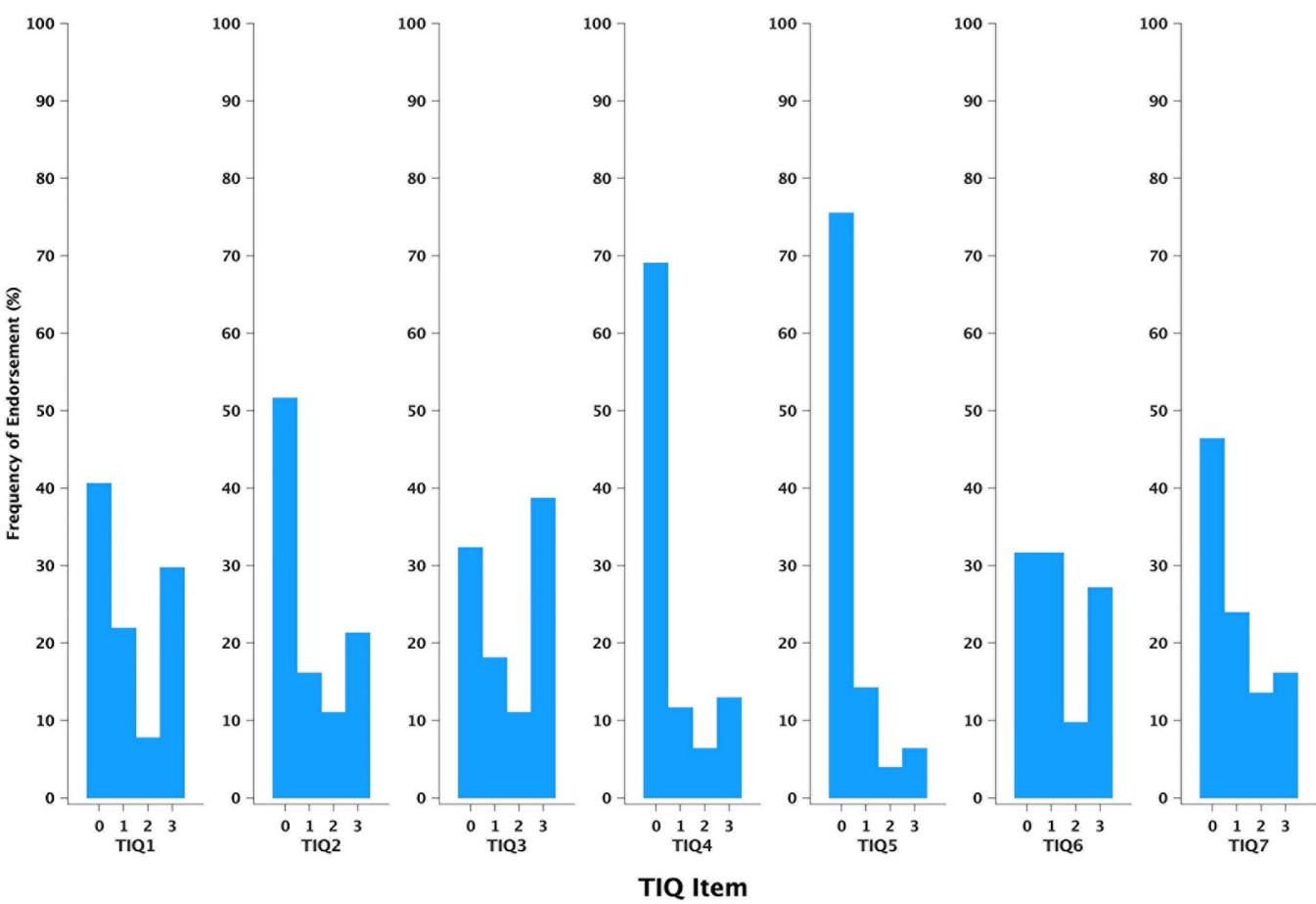

**Fig 1. Histograms of the frequency of endorsement (%).** The histogram shows the frequency of endorsement (%) for each value of each Tinnitus Impact Questionnaire (TIQ) item. Each bar represents the percentage of endorsement frequency (%) for a given response to a given TIQ item (n = 155).

### Discriminant and convergent validity of tinnitus impact categories

The correlations in Table 8 illustrate the convergent validity of the TIQ ordinal categories. There was a large increase in anxiety and depression as measured via the SAD-T and an increase in the VAS of effect of tinnitus on life going from no tinnitus impact to the higher categories of tinnitus impact. Similar results but to a lesser extent were observed for scores for the HIQ, SSSQ, and VAS of tinnitus annoyance and loudness. However, measures of hearing thresholds did not change systematically across the four TIQ ordinal categories, demonstrating the discriminant validity of these categories. Post-hoc analysis using Bonferroni correction indicated that the level of anxiety and depression measured by the SAD-T was significantly lower for patients with no tinnitus impact than for groups with moderate impact ($p = 0.01$) and severe impact ($p = 0.0005$). Additionally, the group with no tinnitus impact had a significantly lower impact of hyperacusis as measured by the HIQ than the group with a severe impact of tinnitus ($p = 0.005$). The group with no impact of tinnitus had significantly lower SSSQ scores than the group with a severe impact of tinnitus ($p = 0.005$). The group with no impact of tinnitus had significantly lower tinnitus annoyance, loudness, and effect on life, quantified by VAS scores, than the group with a severe impact of tinnitus ($p = 0.0005$). VAS scores also differed between the groups with no and moderate tinnitus impact, but only for loudness ($p = 0.03$) and impact on life ($p = 0.05$).

**Table 7. Correlations between TIQ total scores and item scores with age, gender, scores for the SAD-T, SSSQ, HIQ, VAS annoyance, loudness, effect on life, and PTA for the better ears, worse ears and across ears.**

| Scales | TIQ (total) | TIQ 1 | TIQ 2 | TIQ 3 | TIQ 4 | TIQ 5 | TIQ 6 | TIQ 7 |
|---|---|---|---|---|---|---|---|---|
| **Age (n = 155)** | -0.04 | -0.13 | 0.1 | -0.1 | 0.08 | -0.04 | -0.12 | 0.02 |
| **Gender (n = 155)** | 0.02 | -0.06 | 0.07 | -0.03 | 0.08 | 0.05 | 0.02 | 0.02 |
| **SAD-T (n = 118) [a]** | **0.70***** | **0.52***** | **0.63***** | **0.20*** | **0.63***** | **0.36***** | **0.46***** | **0.70***** |
| **HIQ (n = 134) [b]** | **0.30***** | **0.23**** | **0.27**** | -0.02 | **0.26**** | **0.32***** | **0.28***** | **0.25**** |
| **SSSQ (n = 148) [c]** | **0.30***** | **0.17*** | **0.29***** | -0.02 | **0.19*** | **0.3***** | **0.27***** | **0.27**** |
| **VAS (A) (n = 154) [d]** | **0.38***** | **0.37***** | **0.27***** | **0.16*** | **0.33***** | **0.29***** | **0.21**** | **0.32***** |
| **VAS (L) (n = 154) [e]** | **0.43***** | **0.32***** | **0.31***** | **0.35***** | **0.28***** | **0.17*** | **0.31***** | **0.36***** |
| **VAS (E) (n = 154) [f]** | **0.6***** | **0.5***** | **0.5***** | **0.33***** | **0.46***** | **0.36***** | **0.38***** | **0.51***** |
| **PTA (b) (n = 73) [g]** | 0.10 | 0.04 | 0.18 | 0.10 | 0.06 | -0.04 | -0.01 | 0.10 |
| **PTA (w) (n = 73) [h]** | 0.04 | -0.03 | 0.14 | 0.06 | 0.05 | 0.01 | -0.06 | 0.00 |
| **PTA (a) (n = 73) [i]** | 0.07 | -0.00 | 0.17 | 0.08 | 0.06 | -0.01 | -0.05 | 0.04 |

[a] Screening for Anxiety and Depression.

[b] Hyperacusis Impact Questionnaire.

[c] Sound Sensitivity Symptoms Questionnaire.

[d] Visual Analogue Scale Annoyance.

[e] Visual Analogue Scale Loudness.

[f] Visual Analogue Scale Effect on life.

[g] Pure Tone audiometry better ear.

[h] Pure Tone audiometry worse ear.

[i] Pure Tone audiometry across ears.

*p <0.05

**0.01

***0.001.

## Measurement invariance

The MIMIC analysis revealed no direct effects of age, gender or hearing status on the scores for any of the TIQ items. In other words, the seven-item TIQ demonstrated effective measurement invariance with respect to age, gender and hearing status.

## Discussion

This study had a retrospective design. Hence, the analysis was limited to the available clinical data. Not all patients completed the TIQ as a part of their routine care due to time constraints in busy audiology clinics. The response rate was 77.5%, which is satisfactory compared to other studies conducted by primary health care services in the UK National Health Service [54]. To assess the risk of selection bias [55], the characteristics of patients who completed the TIQ were compared with the characteristics of those who did not complete the TIQ. There were no significant differences between responders and non-responders in any of the measures obtained during their standard evaluation for tinnitus and hyperacusis, except for small differences in the VAS for tinnitus annoyance (the average was approximately 6.4 for responders vs 5.9 for non-responders, $p<0.05$), HIQ (2.9 for responders vs 0.9 for non-responders, $p<0.05$), and PTA of the worse ears (the average was approximately 26 dB HL for responders and 18 dB HL for non-responders, $p<0.001$). Therefore, the responders may not be entirely representative of the study population and the results should be interpreted in the light of this limitation. A larger sample with a smaller non-response rate might result in a different outcome.

**Table 8. Comparison of means (SDs) for age, SAD-T, SSSQ, HIQ, VAS annoyance, loudness, effect on life, PTA for better ears, worse ears and across ears for different categories of tinnitus impact based on TIQ scores.**

| | TIQ <5 (no impact) | TIQ 5, 6 (Mild impact) | TIQ 7, 8 (Moderate impact) | TIQ>9 (Severe impact) | $\chi^2(2)$ |
|---|---|---|---|---|---|
| **Age** | 55 (14) n = 64 | 53(8) n = 18 | 57 (16) n = 14 | 534 (16) n = 59 | 1.13 |
| **SAD-T** [a] | 1.4 (2.4) n = 53 | 3.8 (4.3) n = 13 | 4.4 (2.2) n = 10 | 6.7 (3.8) n = 42 | **48.2***** |
| **HIQ** [b] | 1.5 (3.7) n = 58 | 1.8 (2.5) n = 12 | 2.8 (4. 8) n = 13 | 4.7 (6.1) n = 51 | **11.2*** |
| **SSSQ** [c] | 1.2 (2.1) n = 60 | 1.7 (3.7) n = 17 | 1.1 (1.8) n = 14 | 3 (3.5) n = 57 | **13.4**** |
| **VAS (A)** [d] | 6.1 (1.9) n = 64 | 6.2 (1.8) n = 18 | 6.5 (1.8) n = 14 | 7.4 (1.5) n = 58 | **15.9**** |
| **VAS (L)** [e] | 5.4 (2.6) n = 64 | 6.5 (1.7) n = 18 | 7.6 (1.6) n = 14 | 7.7 (2.5) n = 58 | **34.9***** |
| **VAS (E)** [f] | 4.4 (2.8) n = 64 | 5.0 (1.6) n = 18 | 6.4 (2.6) n = 14 | 7.9 (1.6) n = 58 | **56.9***** |
| **PTA (b)** [g] | 15.7 (9) n = 30 | 16.4 (3) n = 7 | 21.2 (13) n = 9 | 18.6 (12) n = 27 | 1.3 |
| **PTA (w)** [h] | 25.0 (18) n = 30 | 23.6 (12) n = 7 | 34.3 (16) n = 9 | 24.9 (19) n = 27 | 4.4 |
| **PTA (a)** [i] | 20.3 (12) n = 30 | 20 (6) n = 7 | 27.8 (11) n = 9 | 21.7 (15) n = 27 | 3.39 |

[a] Screening for Anxiety and Depression.

[b] Hyperacusis Impact Questionnaire.

[c] Sound Sensitivity Symptoms Questionnaire.

[d] Visual Analogue Scale Annoyance.

[e] Visual Analogue Scale Loudness.

[f] Visual Analogue Scale Effect on life.

[g] Pure Tone audiometry better ear.

[h] Pure Tone audioametry worse ear.

[i] Pure Tone audiometry across ears.

*p <0.05

**0.01

***a0.001.

The CFAs indicated that a one-factor solution for the TIQ gave an inadequate fit, with TIQ3 and TIQ5 having low loadings. The low factor loadings suggest that these items did not contribute substantially to the overall measurement of the latent construct. The two-factor model resulted in a worse fit than for the one-factor model. The bi-factor model resulted in a better fit. The RMSEA showed a slight improvement but did not reach the conventional threshold of less than 0.05. However, the other fit indices, including the SRMR, CFI and TFI, had values well within the acceptable range. In a bi-factor model, all items are considered as sufficiently unidimensional, but some items have a degree of construct-relevant multidimensionality that does not interfere with the interpretation of the total score as a one-dimensional measure of the construct [34,56]. The bi-factor model for the TIQ comprises a general factor for tinnitus impact based on all items and specific factors unique to TIQ3 (sleep difficulties) and TIQ5 (inability to perform certain day-to-day activities/tasks). These two items seem to capture functional effects of tinnitus, while the other items seem mainly to assess emotional effects. However, the functional effects of tinnitus are likely to result from the emotional impact of tinnitus (an indirect impact) as opposed to being a direct result of the tinnitus [57]. For example, it is plausible that the emotional distress caused by tinnitus leads to sleep difficulties [58] and the disturbed sleep can lead to increased tinnitus distress, making it harder to carry out day-to-day activities [59]. The idea that sleep difficulties are not a direct effect of tinnitus was explored by Aazh and Moore [58], who reported a mediation analysis showing that the relationship between tinnitus loudness as measured using the VAS and insomnia measured using the insomnia severity index [60] was fully mediated via depression, tinnitus annoyance, and other factors; the direct link between tinnitus loudness and insomnia was non-significant.

To sum up, the bi-factor model that gave the best fit to the data analysed here suggests a possible distinction between direct and indirect effects of tinnitus. This needs to be investigated in future research.

The TIQ demonstrated good internal consistency, with Cronbach's α = 0.84. The only item that if deleted would lead to a slight improvement in Cronbach's α was TIQ3 (Table 5). This item asks about the effect of tinnitus on sleep and scores for TIQ3 were poorly correlated with scores for the other items. However, as shown in Fig 1, TIQ3 exhibited the highest endorsement frequency for response option 3. This means that the majority of patients in our study population reported tinnitus-induced sleep problems, leading to a small range of scores for TIQ3, which could have caused the low correlations between TIQ3 scores and scores for the other items. Sleep problems are among the most frequent complaints of patients with tinnitus [61]. Therefore, we decided that this item should not be deleted. Future studies should use a tinnitus sample from the general population as opposed to patients who seek help from a specialist tinnitus centre. This may lead to a wider range of responses for item TIQ3 and consequently higher correlations between TIQ3 scores and scores for other TIQ items.

There were only very weak and non-significant correlations between TIQ scores and hearing threshold measures. Furthermore, based on the MIMIC analysis, scores for the TIQ items were measurement invariant with respect to hearing status. This is a very important property of the TIQ, as it gives researchers and clinicians a brief tool that is specific to assessing the impact of tinnitus free from any effects of comorbid hearing impairment. In contrast, the scores for many tools for assessing tinnitus are significantly correlated with degree of hearing loss. For example, Gos, Sagan [62] reported that several items of the THI are not measurement invariant with respect to hearing status, which means that the construct being measured differs across groups of people with different levels of hearing loss. Surr, Kolb [63] reported that the THI can be used for assessing the benefit of hearing aids for new hearing aid users with tinnitus. The likely reason for this is that some items of the THI assess hearing problems rather than the impact of tinnitus. This problem is not limited to the THI. Mahafza, Zhao [64] reported that tinnitus severity as measured via the TFI was significantly greater for tinnitus patients with hearing loss than for tinnitus patients with normal hearing thresholds. This may be related to the fact that some items of the TFI assess the impact of hearing loss or to the finding that the severity of tinnitus tends to increase as hearing loss becomes more severe [12,65]. To tease out such effects, it is necessary to assess the impact of tinnitus with tools that are not influenced by hearing impairment, such as the TIQ. Consistent with this argument, it has been suggested that the auditory subscale of the TFI should be excluded when calculating its total score [66,67].

It is very important to use tools that assess the intended construct; otherwise the scores could be misleading. For example, Noguchi, Suzuki [68] reported that the overall TFI score of 21 tinnitus patients decreased significantly after 12 months of using hearing aids. They concluded that hearing aids are highly effective for tinnitus management. However, the observed improvement in their study is likely to be a result of the TFI being sensitive to improvements in hearing problems. Although it has long been established that hearing aids can improve problems related to hearing [69], the evidence base for their effect on tinnitus management is poor [70–72]. To assess whether hearing aids are useful for the management of tinnitus, researchers need to consider using tools such as the TIQ that are not influenced by improvements in hearing.

There was a weak but significant correlation between total TIQ scores and the VAS for tinnitus loudness, even though the TIQ does not include any items asking about the loudness of tinnitus. This may reflect an indirect effect. Tinnitus loudness as measured via the VAS also seems to indirectly predict the severity of anxiety and depression as measured via the hospital

anxiety and depression scale [59,73]. Also, anxiety and depression as measured via the generalised anxiety disorder (GAD-7) questionnaire [74] and patient health questionnaire (PHQ-9) [75] predict the impact of tinnitus as measured via the THI [76]. In the present study, anxiety and depression as measured via the SAD-T were strongly associated with TIQ scores. Taking these effects together, the observed correlation between VAS for tinnitus loudness and TIQ scores is likely to reflect an indirect association between tinnitus loudness and anxiety and depression.

TIQ scores were weakly but significantly correlated with HIQ scores. Similar to tinnitus, the impact of hyperacusis is strongly related to anxiety and depression [77]. This can explain the observed correlation between TIQ and HIQ scores found in this study. While it is also possible that the presence of hyperacusis influences the way that tinnitus affects the patient's life [78,79], a direct link between hyperacusis and the impact of tinnitus seems unlikely. Nevertheless, there is a need for future studies to further explore whether the experience and/or impact of tinnitus is different between patients with tinnitus combined with hyperacusis and those with tinnitus alone.

TIQ scores in the ranges 0–5, 5–6, 7–8, and 9–21 represent no impact, and mild, moderate, and severe tinnitus impact, respectively. There was a clear progressive increase in symptoms of anxiety and depression as measured via the SAD-T going through these categories. SAD-T scores were significantly lower for patients with no tinnitus impact than for groups with moderate tinnitus and severe tinnitus. However, for several measures there was no significant difference between scores for the groups with mild and moderate impact of tinnitus, as shown in Table 8. This could be because only a small proportion of patients fell into those groups. About 12% of patients fell in the mild group and 9% in the moderate group. In other studies using similar patient populations, when the THI was used instead of the TIQ, between 20% and 35% of patients were classified as having a mild or moderate tinnitus handicap [12,44]. Future studies should explore if the cut off values for the TIQ for mild and moderate tinnitus impact need to be adjusted. Interestingly, despite the wider range of TIQ scores for the severe tinnitus impact group in this study, the proportion of patients who were classified as having a severe tinnitus impact via the TIQ was about 38%, which is comparable to the 33% to 44% of patients who were classified as having severe tinnitus handicap when the THI was used with similar populations [12,44]. Since the TIQ ordinal categories can be useful in guiding patient care and for quantifying the impact of tinnitus for medico-legal purposes, future investigations should examine if improvements in the cut off points can be made. It is also important to assess the responsiveness of the TIQ to various tinnitus interventions and to assess what change in the total score can be considered as clinically meaningful.

## Conclusions

The TIQ showed good internal consistency, demonstrated by α and ω values ≥ 0.84. Convergent validity was demonstrated by moderate correlations between TIQ scores and scores for the SAD-T and VAS of tinnitus effect on life. Discriminant validity was demonstrated by very weak and non-significant correlations between TIQ scores and average hearing thresholds and weak correlations with scores for the SSSQ, HIQ, and VAS for tinnitus annoyance and tinnitus loudness. Factor analysis showed that a bi-factor model fitted the data best, with sufficient unidimensionality to support the use of the single TIQ score for assessing the impact of tinnitus. Clear progressive increases in symptoms of anxiety and depression and VAS for effect of tinnitus on life were observed with increasing category of tinnitus impact.

The TIQ provides a potential tool for assessing treatments for tinnitus, as TIQ scores are not influenced by hearing thresholds.

The bi-factor model suggests a possible differentiation between direct and indirect effects of tinnitus. Future studies should explore this possibility in more detail.

## Acknowledgments

We thank two reviewers for helpful comments on an earlier version of this paper.

## Author Contributions

**Conceptualization:** Hashir Aazh.

**Data curation:** Hashir Aazh.

**Formal analysis:** Mercede Erfanian.

**Investigation:** Hashir Aazh.

**Methodology:** Hashir Aazh, Mercede Erfanian.

**Supervision:** Hashir Aazh.

**Writing – original draft:** Hashir Aazh, Mercede Erfanian.

**Writing – review & editing:** Hashir Aazh, Brian C. J. Moore, Mercede Erfanian.

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
