## [Decision Letter · Decision Letter 0]

20 Mar 2024

PONE-D-24-00934Confirmatory factor analysis of the tinnitus impact questionnaire using data from patients seeking help for tinnitus alone or tinnitus combined with hyperacusisPLOS ONE

Dear Dr. Erfanian,

Thank you for submitting your manuscript to PLOS ONE. After careful consideration, we feel that it has merit but does not fully meet PLOS ONE’s publication criteria as it currently stands. Therefore, we invite you to submit a revised version of the manuscript that addresses the points raised during the review process.

We look forward to receiving your revised manuscript.

Kind regards,

Rohit Ravi, Ph.D.

Academic Editor

PLOS ONE

2. For studies involving third-party data, we encourage authors to share any data specific to their analyses that they can legally distribute. PLOS recognizes, however, that authors may be using third-party data they do not have the rights to share. When third-party data cannot be publicly shared, authors must provide all information necessary for interested researchers to apply to gain access to the data. (https://journals.plos.org/plosone/s/data-availability#loc-acceptable-data-access-restrictions)

a) A description of the data set and the third-party source

b) If applicable, verification of permission to use the data set

c) Confirmation of whether the authors received any special privileges in accessing the data that other researchers would not have

d) All necessary contact information others would need to apply to gain access to the data

Additional Editor Comments:

Dear Authors

The paper is well written, I have received the reports of the reviewers and one of the reviewers have suggested minor changes, do incorporate the changes and submit.

Reviewers' comments:

Reviewer's Responses to Questions

**Comments to the Author**

1. Is the manuscript technically sound, and do the data support the conclusions?

Reviewer #1: Yes

Reviewer #2: Yes

2. Has the statistical analysis been performed appropriately and rigorously? 

Reviewer #1: Yes

Reviewer #2: Yes

3. Have the authors made all data underlying the findings in their manuscript fully available?

Reviewer #1: No

Reviewer #2: Yes

4. Is the manuscript presented in an intelligible fashion and written in standard English?

Reviewer #1: Yes

Reviewer #2: Yes

5. Review Comments to the Author

Reviewer #1: In this paper authors assessed factor structure, internal reliability or consistency, discriminant validity, and convergent validity of the modified version of tinnitus impact questionnaire. Method used for the present study was suitable, and conclusions based on the results are appropriate.

Reviewer #2: The m/s describes further validation work on a recently developed tool that seeks to assess the impact of tinnitus on the everyday life of patients with this disorder. The questionnaire has been previously described - this research project considered the internal consistency of the tool and its convergent validity. The latter was considered in relation to the Visual Analogue Scale (effect of tinnitus on life) and the Screening for Anxiety and Depression - Tinnitus questionnaire. Further work using a MIMC model noted no influence of audiometric characteristics, age or gender on the questionnaire results. Overall, this is a very well-written m/s. The flow of ideas was logical and the discussion/conclusions well-grounded in the research findings. A very worthwhile contribution to the research field. My minor comments follow:

Title - Tinnitus Impact Questionnaire should be capitalised.

Text - capitalise Visual Analogue Scale; not necessary to calitalise "Autistic Spectrum Disorders" p. 7 and p. 15); p.9 "... on the number of days it occurred: ..." - specify that the number of days is over a 2 week period; p. 12 subtitle "Discriminant and convergent and validity" needs correction; give a reference for the 'lavaan' package and briefly describe it in the text (an extra phrase will suffice); p. 23 delete comma in "Although, it has long been ..."

References - 1. No need for "The"; 5. should be "HNO"; 7. Should be "Tinnitus Functional Index"; 11. delete "/the British Psychological Society"; 13. journal title change to "Journal of Neuroscience"22. typo ". . "; 25 no capital needed for "People"; 32. typo "; 1997", give report number; 41. "1, editor" and ".;" typos; 44. capitalise "Tinnitus Handicap Inventory"; 45. full stop needed after title; 47. typo "5th ed ed."; location of publisher needed; 61. "The" not required in title; 64. should be " .. Tinnitus Functional Index ..."; 67. volume and page numbers absent; 73. capitalise test name. In general, check all references and be consistent in how titles are presented (some are in full, others are abbreviated - use the PLOS ONE recommended style).

Figures - 1. "TIQ" in full in figure title when first used.

6. PLOS authors have the option to publish the peer review history of their article (what does this mean?). If published, this will include your full peer review and any attached files.

Reviewer #1: No

Reviewer #2: No

---

## [Author Response · Author response to Decision Letter 0]

27 Mar 2024

Dear Dr. Ravi,

We would like to express our gratitude for the opportunity to submit our revised manuscript titled " Confirmatory factor analysis of the tinnitus impact questionnaire using data from patients seeking help for tinnitus alone or tinnitus combined with hyperacusis" to PLOS ONE for consideration. We appreciate the constructive feedback provided by the reviewers and the editor, which we have considered in revising our manuscript.

In response to the reviewers' comments, we have made revisions to address the concerns raised, and these are highlighted in yellow in the text. Specifically:

1. Reviewer 1:

We thank reviewer #1 for their time to review the manuscript and positive feedback.

2. Reviewer 2:

Comment 1) Title - Tinnitus Impact Questionnaire should be capitalised.

Response 1): We have amended the title as requested.

Comment 2) Text - capitalise Visual Analogue Scale; not necessary to capitalise "Autistic Spectrum Disorders".

Response 2) The term "Visual Analogue Scale" has been capitalised and all instances of "Autistic Spectrum Disorders" have been converted to lower.

 Comment 3) p. 7 and p. 15); p.9 "... on the number of days it occurred: ..." - specify that the number of days is over a 2 week period; 

Response 3) Changed as suggested.

Comment 4) p. 12 subtitle "Discriminant and convergent and validity" needs correction; 

Response 4) The text has been corrected.

Comment 5) give a reference for the 'lavaan' package and briefly describe it in the text (an extra phrase will suffice);

Response 5) A reference has been added, and a brief description has been added.

 Comment 6) p. 23 delete comma in "Although, it has long been ..."

Response 6) Comma has been removed.

Comment 7) References - 1. No need for "The"; 5. should be "HNO"; 7. Should be "Tinnitus Functional Index"; 11. delete "/the British Psychological Society"; 13. journal title change to "Journal of Neuroscience"22. typo ". . "; 25 no capital needed for "People"; 32. typo "; 1997", give report number; 41. "1, editor" and ".;" typos; 44. capitalise "Tinnitus Handicap Inventory"; 45. full stop needed after title; 47. typo "5th ed ed."; location of publisher needed; 61. "The" not required in title; 64. should be " .. Tinnitus Functional Index ..."; 67. volume and page numbers absent; 73. capitalise test name. In general, check all references and be consistent in how titles are presented (some are in full, others are abbreviated - use the PLOS ONE recommended style).

Response 7) We have downloaded the PLOS ONE EndNote citation style and updated the reference list accordingly.

Sincerely,

Mercede Erfanian (PhD)

---

## [Editor Report · Decision Letter 1]

12 Apr 2024

Confirmatory factor analysis of the Tinnitus Impact Questionnaire using data from patients seeking help for tinnitus alone or tinnitus combined with hyperacusis

PONE-D-24-00934R1

Dear Dr. Erfanian,

We’re pleased to inform you that your manuscript has been judged scientifically suitable for publication and will be formally accepted for publication once it meets all outstanding technical requirements.

Kind regards,

Rohit Ravi, Ph.D.

Academic Editor

PLOS ONE

Additional Editor Comments (optional):

Dear Authors, the revision is satisfactory.

Reviewers' comments:

NIL

---

## [Editor Report · Acceptance letter]

26 Apr 2024

PONE-D-24-00934R1 

PLOS ONE

Dear Dr. Erfanian, 

I'm pleased to inform you that your manuscript has been deemed suitable for publication in PLOS ONE. Congratulations! Your manuscript is now being handed over to our production team.

Kind regards, 

on behalf of

Dr. Rohit Ravi 

Academic Editor

PLOS ONE